# Rasch Model of the COVID-19 Symptom Checklist—A Psychometric Validation Study

**DOI:** 10.3390/v13091762

**Published:** 2021-09-03

**Authors:** Tanja A. Stamm, Valentin Ritschl, Maisa Omara, Margaret R. Andrews, Nils Mevenkamp, Angelika Rzepka, Michael Schirmer, Siegfried Walch, Thomas Salzberger, Erika Mosor

**Affiliations:** 1Section for Outcomes Research, Center for Medical Statistics, Informatics and Intelligent Systems, Medical University of Vienna, Spitalgasse 23, 1090 Vienna, Austria; valentin.ritschl@meduniwien.ac.at (V.R.); maisa.omara@meduniwien.ac.at (M.O.); margaret.andrews@meduniwien.ac.at (M.R.A.); erika.mosor@meduniwien.ac.at (E.M.); 2Ludwig Boltzmann Institute for Arthritis and Rehabilitation, 1090 Vienna, Austria; 3Center for Social- & Health Innovation, MCI—The Entrepreneurial School, Universitätsstraße 15, 6020 Innsbruck, Austria; Nils.Mevenkamp@mci.edu (N.M.); Siegfried.Walch@mci.edu (S.W.); 4Center for Health & Bioresources, AIT Austrian Institute of Technology GmbH, Digital Health Information Systems, Reininghausstrasse 13/1, 8020 Graz, Austria; Angelika.Rzepka@ait.ac.at; 5Department of Internal Medicine, Clinic II, Medical University of Innsbruck, Anichstrasse 35, 6020 Innsbruck, Austria; michael.schirmer@i-med.ac.at; 6Institute for Statistics and Mathematics, University of Economics and Business of Vienna, Welthandelsplatz 1, 1020 Vienna, Austria; thomas.salzberger@gmail.com

**Keywords:** coronavirus, COVID-19, SARS CoV-2, symptom checklist, self-reported instrument, construct validity

## Abstract

While self-reported Coronavirus Disease 2019 (COVID-19) symptom checklists have been extensively used during the pandemic, they have not been sufficiently validated from a psychometric perspective. We, therefore, used advanced psychometric modelling to explore the construct validity and internal consistency of an online self-reported COVID-19 symptom checklist and suggested adaptations where necessary. Fit to the Rasch model was examined in a sample of 1638 Austrian citizens who completed the checklist on up to 20 days during a lockdown. The items’ fatigue’, ‘headache’ and ‘sneezing’ had the highest likelihood to be affirmed. The longitudinal application of the symptom checklist increased the fit to the Rasch model. The item ‘cough’ showed a significant misfit to the fundamental measurement model and an additional dependency to ‘dry cough/no sputum production’. Several personal factors, such as gender, age group, educational status, COVID-19 test status, comorbidities, immunosuppressive medication, pregnancy and pollen allergy led to systematic differences in the patterns of how symptoms were affirmed. Raw scores’ adjustments ranged from ±0.01 to ±0.25 on the metric scales (0 to 10). Except for some basic adaptations that increases the scale’s construct validity and internal consistency, the present analysis supports the combination of items. More accurate item wordings co-created with laypersons would lead to a common understanding of what is meant by a specific symptom. Adjustments for personal factors and comorbidities would allow for better clinical interpretations of self-reported symptom data.

## 1. Introduction

The novel Coronavirus Disease 2019 (COVID-19) spread rapidly worldwide, and the number of cases increased globally at an accelerated rate [1,2]. While measures are needed to decrease the virus spreading and mitigate the impact of the pandemic, effective monitoring is essential to tailor these measures to the current situation. Self-reported symptom tracking contributes to monitoring [3], and several self-reported symptom checklists and trackers for COVID-19 exist; some also allow their users to estimate the risk for a Severe Acute Respiratory Syndrome Coronavirus-2 (SARS-CoV-2) infection [4]. A self-reported symptom checklist refers to a questionnaire where the participants themselves indicate whether or not they experience a specific symptom without interference. Moreover, self-reported symptom-tracking could increase safety in COVID-19 patients at home who do not need hospitalisation. In oncology, even survival could be improved by collecting self-reported side-effects on time [5]. Thus, self-reported data are a vital part of a comprehensive outcome measurement scheme, and large international research initiatives are developing the technical, ethical and legal infrastructure to collect such data at the country-level. While self-reported COVID-19 symptom checklists have been extensively used during the pandemic, we did not find any study that validated such a checklist from a psychometric perspective, except for one Italian study that aimed to identify people with COVID-19 by using self-reported symptoms and also determined the factorial structure of the symptom checklist, but did not perform any other psychometric testing [6].

Self-reported patient data are acquired with specific instruments. To be fit-for-purpose, these tools must fulfil certain psychometric properties. Most commonly, psychometric properties are examined based on the classical test theory [7,8,9,10]. However, the classical test theory focuses on overall, sample-based statistics, which provide little insight into how individual items actually work [10]. Furthermore, the classical test theory makes specific assumptions, such as normally distributed populations and interval-scaled response data, that are rarely met in practice. An alternative psychometric approach is the Rasch model [11]. It overcomes some of the simplistic assumptions of the classical test theory and provides insights at the item level. A range of possible misfits to the Rasch model exist. A basic test compares actual responses to the expected responses based on parameter estimates. Another test checks if the expected response to a specific item differs between respondents from different sub-groups of persons (e.g., females versus males, people from different age groups, or people with versus without comorbidities), but has the same ‘ability’/likelihood to affirm a certain number of items. This is referred to as differential item functioning [12]. If differential item functioning is apparent, parameter invariance is violated, and findings between different groups of people cannot be compared without adjusting the scale. Another fundamental requirement of measurement is unidimensionality [13], which applies whenever a scale is summarized by one measure.

Although a variety of self-reported COVID-19 symptom screening checklists exist [4], items are often not formulated in a standard manner, and psychometric information on these instruments is scarce. This lack of knowledge of the psychometric properties of the symptom checklists prevents us from judging whether we use them in the right way or whether adaptations might be necessary. As more symptoms might increase the likelihood for a suspected case, this might be similar to measuring a latent construct. Furthermore, a recent Cochrane review concluded that individual COVID-19 signs and symptoms, such as cough, sore throat, fever, myalgia or arthralgia, fatigue, and headache, had very poor diagnostic power [14], and molecular screening and diagnostic tests are essential. Moreover, inaccurate measurement in self-reported instruments jeopardizes the comparability of the results. We, therefore, used the Rasch model to explore construct validity and internal consistency [15] of an online self-reported COVID-19 symptom checklist and suggested adaptations where necessary. 

## 2. Materials and Methods

### 2.1. Design and Participants

We conducted a psychometric study using a sample of 1638 Austrian citizens who completed an online COVID-19 symptom checklist on up to 20 days during a period of restrictive country-wide COVID-19 measures. After the first confirmed cases of COVID-19 in Austria on 25 February 2020, nationwide infection control measures were ordered by the Austrian government from 16 March 2020 onwards. Public life in Austria remained severely affected before the first easing measures were implemented in mid-April 2020 [16]. Therefore, the self-reported, online COVID-19 symptom checklist used in the present study was available from 22 March to 30 April 2020. Participants were recruited by the organizations to which the co-authors were affiliated via social media and snowball sampling (people were asked to share the link with others). Participants were able to download information about the study and the informed consent form via a link provided and gave written informed consent by ticking a box before filling in the questionnaire. For children below 18 years, we obtained the consent of their parents. There were no specific exclusion criteria defined.

### 2.2. Data Collection

We used the checklist for symptom descriptions for COVID-19 proposed by WHO [17] and included fever, fatigue, cough, dry cough/no sputum production, pain in limbs, sore throat, headache, shortness of breath, chills, vomiting, diarrhea, nasal congestion, sneezing, sniffles/rhinitis and smell and taste disorders (Appendix A). Response options and scores for each item were as follows: “Yes” (scoring 1), “No” (0) and “I can’t say that” (3). For the analysis, we dichotomized the items by collapsing “No” and “I can’t say that” because we assumed that these two answers would indicate that the participant had not experienced a particular symptom. In addition, we asked the participants to state gender, age group, highest completed education, current smoking status, body height and weight, whether any type of COVID-19 test had been conducted and if so, what the result of this test was, whether any comorbidities existed (nervous system, cardiovascular, gastrointestinal, liver, kidney, oncologic, high blood pressure and/or diabetes), whether he or she was taking immunosuppressive medication and whether the participant was pregnant (females only). We used a self-reported code consisting of numbers and strings based on first name initials, initials of first names of relatives and birth months to allocate multiple assessments to the correct participants, despite guaranteeing anonymity. Participants were asked to fill in the symptom checklist on a daily basis. Due to the psychometric nature of this study, only complete cases were included. The relevant ethical committees approved the study (Medical University of Vienna 1379/2020, Vienna, Austria, Medical University of Innsbruck, Innsbruck, Austria, 1076/2020 and ethical committee of the region Vorarlberg, Vorarlberg, Austria).

### 2.3. Statistical Analysis

#### 2.3.1. Fit to the Rasch Measurement Model

Overall and item-based fit to the Rasch model was explored in a series of dichotomous models [18] using two different data sets: one data set with the questionnaire filled in for the first time when the participants entered the study and another data set in which we recorded a symptom to be affirmed by a participant if it was ticked at least once during the period of nationwide restrictive measures. We used raw scores without weighting [19] and estimated item and person location parameters. Based on the Rasch model, items can be brought into a hierarchy according to their level of ‘difficulty’ or likelihood to be affirmed, which means that more ‘difficult’ items are only affirmed by persons with higher ‘abilities’. The item location parameter determines the likelihood of each item to be affirmed. Likewise, a hierarchy of persons can be established based on the likelihood that a person is likely to affirm more or fewer items.

Item fit residuals between −2.5 and +2.5 with non-significant F-tests represented good individual item fit. Non-significant chi-squared values were interpreted as fit to the latent trait. Local dependency between items was determined using residual correlations based on a cut-off of 0.2 above their mean [20]. Local dependency represents an additional dependency between items beyond the relationship associated with the latent construct measured by the instrument. Thus, local dependency also distorts the metric of the measure. We compared Cronbach’s alpha with the person separation index (PSI) to assess the instrument’s item-based internal consistency and reliability. The PSI refers to the reproducibility of relative measure location and indicates whether a scale can distinguish between people with higher and lower levels of the concept measured by the instrument [21]; in general, a PSI ≥ 0.7 indicates that the instrument is sufficiently suitable for group comparisons.

#### 2.3.2. Unidimensionality

Unidimensionality refers to whenever the items can be summarized by one latent construct as measured in a single score. To test unidimensionality, we used an approach proposed by Smith [22] and combined principal component analysis of the item residuals with a series of t-tests to assess whether subsets of residuals that loaded positively or negatively resulted in different estimates of person parameters. These sets of items were chosen to maximize the contrast between them and were thus then most likely to violate the assumption of unidimensionality [23].

#### 2.3.3. Differential Item Functioning

Differential item functioning was assessed separately for each item by comparing the expected responses to a specific item between respondents from different sub-groups who shared the same likelihood to affirm a certain number of items. The sub-groups were built based on gender (female, male, divers/other), age group (10 sub-groups listed in Table 1), highest completed education (6 levels listed in Table 1), COVID-19 test status (pos/neg/no test), comorbidities (yes/no), immunosuppressive medication (yes/no), pregnancy (yes/no), current smoking status (yes/earlier, but not now/never) and body mass index (BMI; above versus below median). As only seven participants indicated ’divers/other’ for gender, we did not include this category into the differential item functioning analysis and compared only female and male participants. If differential item functioning was apparent in an item for a personal factor with more than two properties, we determined between which sub-groups these differences occurred using post hoc analysis of the residual means.

#### 2.3.4. Person-Item Targeting

Person-item targeting indicates whether the items cover the range of persons or whether items are “too easy” or “too difficult” for a certain population. It was inspected graphically using a person-item map.

#### 2.3.5. Transformation to a Metric Interval Scale

Based on the logit scale from the Rasch model, we transformed the raw sum scores into a metric scale by linearly transforming the logits on a scale from 0 to 10. If differential item functioning existed for a personal factor, we split the respective item and performed a separate metric transformation for each sub-group. We used the differences between these metric scales to adjust the scores for the respective personal factor, e.g., for people with and without comorbidities.

#### 2.3.6. Sample Size Considerations

Studies on sample sizes for the Rasch model for polytomous data recommend including up to 1600 cases and showed a significant overfit when larger numbers were used [24]. Larger samples also imply excessive power of the test of fit, flagging trivial discrepancies as statistically significant.

#### 2.3.7. Additional Aspects of the Analyses and Reporting of the Results

All analyses were performed with either Microsoft Excel, RUMM2030 or the eRm and ltm packages in R (www.r-project.org, accessed on 31 July 2021). We used the reporting criteria for psychometric studies according to Streiner and Kottner (2014) as a reference (Appendix A) [25].

## 3. Results

Participants from all age groups, ranging from ≤9 years to ≥90 years, filled in the symptom checklist. Two thirds of the participants (66%) were female (Table 1). Twelve participants (0.7%) indicated fever at least once during the period of nationwide restrictive measures (8 [0.5%] when they first filled in the questionnaire); 945 (57.7%; first data entry: 459; 28%) reported fatigue, 409 (25%; first data entry: 158; 9.6%) cough, 90 (5.5%; first data entry: 76; 4.6%) dry cough/no sputum production, 238 (14.5%; first data entry: 91; 5.6%) pain in limbs, 482 (29.4%; first data entry: 116; 7.1%) sore throat, 786 (48%; first data entry: 203; 12.4%) headache, 181 (11.1%; first data entry: 70; 4.3%) shortness of breath, 98 (6%; first data entry: 14; 0.9%) chills, 133 (8.1%; first data entry: 21; 1.6%) vomiting, 308 (18.8%; first data entry: 74; 4.5%) diarrhea, 604 (36.9%; first data entry: 317; 19.4%) nasal congestion, 752 (45.9%; first data entry: 361; 22%) sneezing, 436 (26.6%; first data entry: 229; 14%) sniffles/rhinitis and 108 (6.6%; first data entry: 81; 4.9%) smell and taste disorders.

### 3.1. Diagnosis of Measurement Problems

The data set with the first responses of the participants (model 1 in Table 2) showed a considerable misfit to the Rasch model and a substantial discrepancy between PSI (−0.06) and Cronbach’s alpha (0.68). The second data set, where a symptom was recorded as affirmed if it was scored at least once during the observation period (model 2 in Table 2), showed a better model fit (a lower Root Mean Square Error of Approximation) and a substantially smaller difference between PSI (0.60) and Cronbach’s alpha (0.75). This indicated that the scoring in the second data set was more appropriate. Due to the better model fit and the fact that fit statistics of symptom checklists could be affected if only a few persons of a population report symptoms, we decided to use only the second data set for further analyses, where a symptom was recorded as affirmed if it was scored at least once during the observation period. In addition, item fit statistics on the first responses of the participants are shown in Appendix A.

Only one item (cough) had significantly deviating F-tests with an item-based fit residual being below −2.5 and a significantly deviating chi-squared value (Table 3). Fatigue, headache and sneezing were the items with the highest likelihood to be affirmed, compared to fever with the lowest probability of affirmation by the participants, followed by dry cough and chills. 

### 3.2. Adjustment for Differential Item Functioning

Except for smoking status and BMI, all personal factors led to differential item functioning in certain items (Table 3). As expected, taste and smell disorders were more likely to be affirmed by participants with a positive COVID-19 test result. Participants with a pollen allergy were more likely to indicate nasal congestion and sneezing, and pregnant women were more likely to experience vomiting. However, less obvious was that participants with comorbidities exhibited a higher likelihood to indicate shortness of breath and were less likely to score a headache when compared to participants without comorbidities; pain in the limbs was more often affirmed by persons with immunosuppressive medication than persons without; women were more likely to indicate headache when compared to men, and men indicated more often rhinitis/sniffles than women (Figure 1). Participants with a higher educational status (completed post-secondary non-tertiary education or first stage of tertiary education) were more likely to indicate fatigue than people with completed apprenticeships (the significant absolute residual mean differences in the post hoc analysis were 0.29 and 0.4, respectively) and unfinished compulsory education (the significant absolute residual mean differences were 0.5 and 0.61, respectively). Likewise, people with a completed second stage of tertiary education were more likely to affirm fatigue than participants with unfinished compulsory education (the significant absolute residual mean difference was 0.5). Fatigue was also most prevalent in younger adults (from 20 to 49 years of age) compared to both children/adolescents up to 19 and older adults of 50+ years (Appendix A). However, as the highest completed education also depended on the age group, e.g., participants below the age of 20 could not have completed the second stage of tertiary education, and the numbers of participants in the age groups were heterogeneous (ranging from 1 to 541; Table 1), we did not further assess the differences between specific age groups. 

The bar charts on the left depict the absolute and relative frequencies of answers regarding an item in persons with and without comorbidities and immunosuppressive medication, as well as gender. While persons with comorbidities were more likely to indicate shortness of breath (87/359; 24%) compared to people without comorbidities (94/1279; 7%), they were less likely to affirm a headache (160/359 (45%) without compared to 626/1279 (49%) with comorbidities). Persons with immunosuppressive medication were more likely to indicate pain in limbs (21/45; 47%) than people without (217/1593; 14%). Women were more likely to indicate a headache (596/1088 (55%) versus 187/534 (34%) in men) and men rhinitis/sniffles (153/534 (28%) versus 280/188 (26%) in women). The item characteristic curves in the right column were split for people with comorbidities/immunosuppressive medication (red curves) and without (blue curves), as well as women (blue curves) and men (red curves).

### 3.3. Local Dependency in Relation to Item Fit

Local dependency was detected between items 3 (cough) and 4 (dry cough/no sputum production), items 12 (nasal congestion) and 13 (sneezing), as well as items 12 (nasal congestion) and 14 (sniffles/rhinitis). The additional ‘local’ dependency of nasal congestion, sneezing and/or sniffles/rhinitis seemed to be rather evident and well known. Moreover, these three symptoms might be less important regarding COVID-19 than, for example, cough. Cough and dry cough also showed such an additional dependency which might also be relate to the misfit of cough (item 3) in Table 3. Finally, we found both data sets to be unidimensional (the last four far-right columns in Table 2 show these results).

### 3.4. Person Item Targeting

From the graphical inspection of the person-item map (Appendix A), it is evident that the items, in general, cover the range of symptoms in the population. We used the data set where an item was recorded as affirmed, if indicated at least once during the observation period, which consisted of fewer zero-scored symptoms than the data set of the questionnaires filled in only once. 

### 3.5. Transformation to a Metric Interval Scale

The raw sum scores and the corresponding values on an interval metric scale from zero to ten are depicted in Table 4. Higher scores indicate more symptoms. We split and adjusted items ‘headache’, ‘shortness of breath’ and ‘rhinitis/sniffles’, ‘pain in limbs’ and ‘fatigue’ according for the personal factor comorbidities, gender, immunosuppressive medication and educational status (Table 4; details of the calculations are depicted in Appendix A). Adjustments for personal factors differed in magnitude over the range of each metric scale.

## 4. Discussion

We assessed construct validity and internal consistency of a COVID-19 symptom checklist using advanced psychometric methods and suggested some basic adaptations where necessary. Construct validity refers to the degree to which a self-reported instrument measures the construct it intends to measure concerning, e.g., relationship to the scores of other instruments or the differences between relevant groups. Internal consistency is a reliability measure and refers to the degree of the interrelatedness among the items [15]. The extent to which instruments fulfil these psychometric properties increases their clinical relevance and the comparability of the results. In addition to other approaches, such as those proposed by the classical test theory, the Rasch model provides further insights into whether and how items should be formulated, whether they cover the full range of persons’ symptom levels and are not “too easy” or “too difficult” for specific populations, how many answer options should be provided per item, if the scale is unidimensional and whether symptom scores could also depend on personal or contextual factors, such as educational levels, gender, age groups, comorbidities or the intake of specific medications. In addition, the Rasch model provides a basis for transforming ordinal data into metric interval scales. Other psychometric models in Item Response Theory take more parameters into account, including discrimination and guessing. However, as symptom checklists are commonly based on comprehensive symptom descriptions and used for monitoring, all items should be kept in the instrument instead of selecting only a few, which best discriminate. Similarly, guessing is more relevant for performance tests than symptom checklists which do not test patients’ abilities but intend to capture patient experiences in daily life. To the best of our knowledge, this is the first psychometric study on such a checklist using the Rasch model, despite these checklists being used extensively during the pandemic [4]. Self-reported questions on whether patients experienced certain bodily symptoms are also included in questionnaires in other medical areas with similar clinical implications: the information coming directly from the patients adds to the understanding of the impact of a disease on patients’ lives and complements the parameters assessed by the clinicians [23]. 

Our analysis also showed certain areas where the checklist could be improved before it can be used in clinical practice. Local dependency between the items ‘cough’ and ‘dry cough/no sputum production’, for example, showed the need for more accurate item wordings, which would make it clearer to the participants how to differentiate between these items. The unclear difference between what is meant by a cough’ and a ‘dry cough/no sputum production’ could have prevented them from precise scoring, and an improved item wording would be preferable. Likewise, internationally agreed standard item wordings for COVID-19 symptom checklists co-created with laypersons could improve the comparability of the findings, especially across data sets and different countries. Such an agreed version should then also be tested for construct validity and internal consistency using an approach similar to ours.

Another issue with clinical relevance which our analysis showed is that participants with certain characteristics, such as comorbidities, intake of immunosuppressive medications, gender and educational status, scored systematically differently on some items, although they had, in principle, the same likelihood to affirm the respective items. This means, for example, that people with comorbidities had a higher likelihood to affirm shortness of breath due to this confounder. Likewise, people with a higher educational status were more likely to score fatigue despite their otherwise similar likelihood to affirm a similar number of items when compared to people with a lower educational status. The symptoms on the COVID-19 checklists could also occur due to other diseases or concerning certain personal characteristics. According to these characteristics, an adjustment of scores would increase the comparability of the findings between individuals from different sub-groups. 

The low PSI in the data set with the questionnaires filled in for the first time by the participants could be related to the small number of people who affirmed the items when they first responded to the questionnaire. Thus, it might be difficult to determine construct validity and internal consistency of symptom checklists if only a small number of people report symptoms. We recorded similar findings in previous studies where a large number of zero-scored symptoms led to right-skewed distributions and an unfavourable person-item targeting [23,26]. In general, the validity of self-reported symptom checklists, especially in the case of easily fluctuating symptoms, such as the ones used in our COVID-19 checklist, might be increased by using these checklists longitudinally.

We could show that fever, dry cough and chills had a low probability of affirmation. This could be related to the fact that participants were anxious to report these symptoms digitally, even if anonymity was guaranteed and participation in our study was voluntary. Participants could be reluctant to fill in such symptoms in digital tools and might fear that this could lead to personal consequences, such as quarantine or other restrictions. This fact needs to be considered when self-reported checklists are used and could be a reason for the very poor diagnostic power [13]. Another relevant study shows that the use of questionnaires will not identify all infected patients and molecular screening and diagnostic tests are necessary [27]. Our study, however, does not suggest using questionnaires instead of other screening procedures, but in addition to them.

As we used Austrian data in our study, differential item functioning between countries and assessing cross-cultural differences could thus not be assessed. Another limitation of our study is that we collected self-reported comorbidities. Future work would be interesting if we were to repeat the Rasch model in an international data set collected by an online self-reported COVID-19 symptom checklist with adapted wording of specific items. Further research is needed with regard to the reliability of the checklist.

## 5. Conclusions

Apart from some basic adjustments that increases the scale’s construct validity and internal consistency, the analysis supports the present combination of items into a comprehensive COVID-19 symptom questionnaire. More accurate item wordings co-created with laypersons would lead to a common understanding of what is meant by a specific symptom. Adjustments for personal factors and comorbidities would allow for better clinical interpretations of self-reported symptom data.

## Figures and Tables

**Figure 1 viruses-13-01762-f001:**
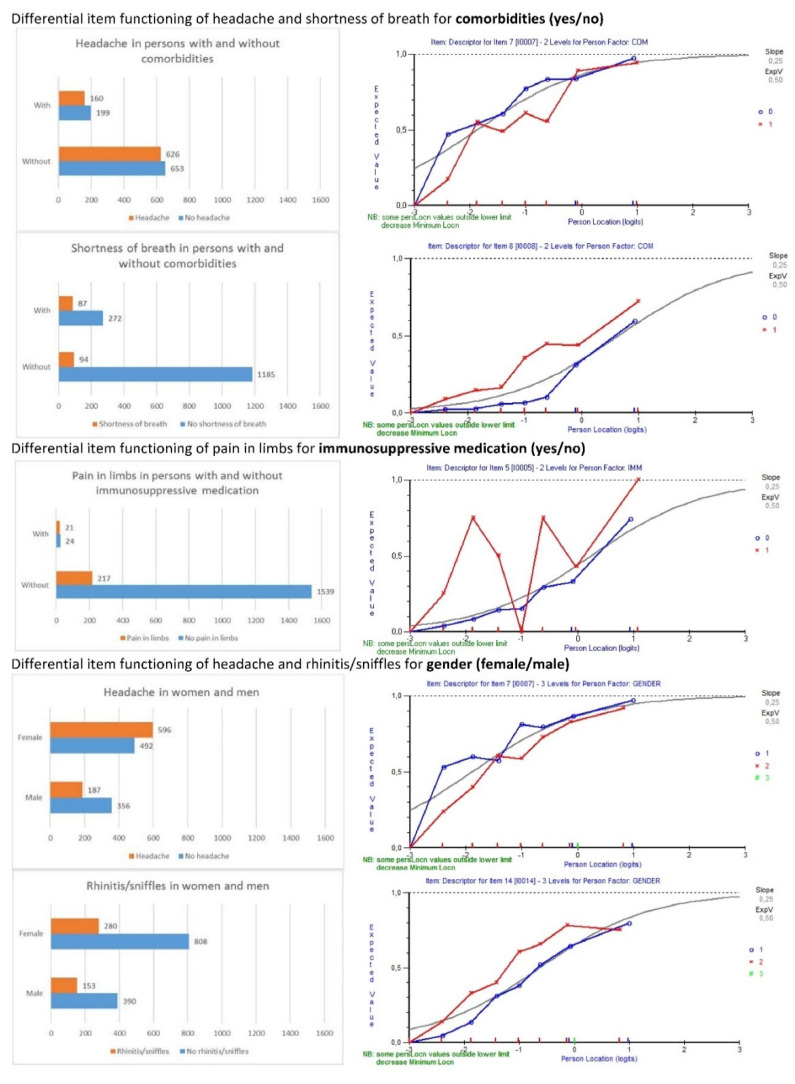
Differential item functioning for comorbidities, immunosuppressive medication and gender.

**Table 1 viruses-13-01762-t001:** Participant characteristics.

Personal Factors/Characteristics	Frequencies
**Total Number of Participants** *N* (%)	1638 (100%)
**Gender** *n* (%)	
Female	1088 (66.4%)
Male	543 (33.2%)
Divers/other	7 (0.4%)
**Pregnancy** (women only) *n* (%)	17 (1.6% of the women)
**Age Groups** *n* (%)	
≤9 Years	21 (1.3%)
10–19 Years	31 (1.9%)
20–29 Years	421 (25.7%)
30–39 Years	451 (27.5%)
40–49 Years	262 (16.0%)
50–59 Years	264 (16.1%)
60–69 Years	137 (8.4%)
70–79 Years	40 (2.4%)
80–89 Years	10 (0.6%)
≥90 Years	1 (0.1%)
**Highest Education** *n* (%)	
Unfinished compulsory education	37 (2.3%)
Completed compulsory education	30 (1.8%)
Completed apprenticeship	140 (8.5%)
Completed post-secondary non-tertiary education	453 (27.7%)
Completed first stage of tertiary education	718 (43.8%)
Completed second stage of tertiary education	260 (15.9%)
**COVID-19 Tested** *n* (%)	
Positive	16 (1%)
Negative	187 (11.4%)
Not tested	1435 (87.6%)
**Comorbidities** *n* (%)	
Yes	359 (21.9%)
No	1279 (78.1%)
**Immunosuppressive Medication** *n* (%)	
Yes	45 (2.7%)
No	1593 (97.3%)

Note. *N* (%) = total number of participants (percentage); *n* (%) number of participants (percentage).

**Table 2 viruses-13-01762-t002:** Model fit statistics. Mean item log residual test of fit, item–trait interaction chi-square statistics, and Root Mean Square Error of Approximation (RMSEA) were calculated to assess model fit. Model 1 (“First”) refers to the questionnaires filled in for the first time when the participants entered the study. In Model 2 (“Ever”), a symptom was considered affirmed by a participant if it was ticked at least once during the period of nationwide restrictive measures.

	Model Fit Statistics							Unidimensionality Analysis			
	Mean Item Location (±SD)	Mean Item Fit Residual (±SD)	Mean Person Location (±SD)	Mean Person Fit Residual (±SD)	Person Separa-tion Index (PSI)	Cron-Bach’s α	Root Mean Square Error of Approx-Imation (RMSEA)	Number of Sign. *t*-tests	Sample	% of Sign. *t*-tests	Lower Bound of 95% CI
**Model 1.** “First”	0 (±1.42)	−1.19 (±2.33)	−2.89 (±1.04)	−0.3 (±0.52)	−0.06	0.68	0.052	14	1638	0.9%	−0.2%
**Model 2.** “Ever”	0 (±1.59)	−0.86 (±1.49)	−1.91 (±1.37)	−0.26 (±0.57)	0.60	0.75	0.027	43	1638	2.6%	1.6%

CI = Confidence interval.

**Table 3 viruses-13-01762-t003:** Item fit statistics sorted in ascending order according to item location in the data set with the items affirmed, if they were scored positively at least once by each participant during the observation period. The smallest (negative) item location of fatigue implies that it was the item with the highest likelihood to be affirmed, whereas fever showed the lowest likelihood. Only one item (item 3 cough) showed a fit residual below the threshold of −2.5 and a significant F-test, which represented individual item misfit. Cough also had a significant chi-squared value which can be interpreted as misfit to the latent trait. The Bonferroni corrected significance level was 0.000667. Significances are highlighted in bold letters. Differential item functioning (DIF; depicted in the far right column) exists, if the expected response to a specific item differs between respondents from different sub-groups of persons (e.g., females versus males, people from different age groups or people with and without comorbidities), even though they have the same ‘ability’/likelihood to affirm the same number of items.

Item Number	Description	Location	Standard Error	Fit Residual	F-Stat	*p*-Value	ChiSq	*p*-Value	DIF for
2	Fatigue	−2.47	0.06	−0.47	0.41	0.9284	4.45	0.7271	Age; educational status
7	Headache	−1.87	0.06	0.11	1.70	0.0839	14.29	0.0463	Gender; comorbidities
13	Sneezing	−1.75	0.06	1.28	2.02	0.0342	17.95	0.0122	Pollen allergy
12	Nasal congestion	−1.26	0.06	−1.96	2.63	0.0051	20.61	0.0044	Pollen allergy
6	Sore throat	−0.84	0.06	0.11	0.68	0.7264	7.23	0.4056	-
14	Sniffles/rhinitis	−0.62	0.07	−2.59	2.96	0.0017	23.54	0.0014	Gender
3	Cough	−0.60	0.07	−3.55	4.73	**0.0000**	38.43	**0.0000**	-
11	Diarrhea	−0.15	0.07	1.61	1.38	0.1928	15.32	0.0322	-
5	Pain in limbs	0.24	0.08	−1.55	1.78	0.0667	17.71	0.0134	Age
8	Shortness of breath	0.65	0.08	−2.37	1.24	0.2688	13.35	0.0641	Comorbidities
10	Vomiting	1.01	0.09	−1.41	1.17	0.3094	12.41	0.0879	Pregnancy
15	Smell and taste disorders	1.21	0.10	0.47	1.33	0.2188	14.21	0.0475	COVID-19 test
9	Chills	1.44	0.11	−1.73	0.92	0.5077	9.67	0.2081	-
4	Dry cough/no sputum production	1.47	0.11	−1.10	1.16	0.3164	11.26	0.1275	Age
1	Fever	3.55	0.25	0.24	0.98	0.4555	10.44	0.1652	

**Table 4 viruses-13-01762-t004:** Transformations of raw scores into metric scales (logit scales and from zero to ten). ‘COM’ refers to persons with comorbidities; ‘IMM’ refers to persons with immunosuppressive medication; the numbers below COM, IMM and gender represent the difference between the metric scales for these subgroups, e.g., persons with and without comorbidities, and represent a possibility to adjust the metric scores for a better comparability of persons with different personal factors.

All—O Split			COM		IMM		Gender		Education Level	
Raw Score	Metric Scale	Zero to Ten	With	Without	With	Without	Female	Male	Lower Than Post Secondary	≥Post Secondary
0	−4.02	0.55	0.06	−0.06	−0.08	0.08	−0.08	0.08	0.25	−0.25
1	−3.10	1.56	0.06	−0.06	−0.09	0.09	−0.07	0.07	0.19	−0.19
2	−2.40	2.34	0.04	−0.04	−0.11	0.11	−0.05	0.05	0.15	−0.15
3	−1.87	2.93	0.02	−0.02	−0.13	0.13	−0.02	0.02	0.11	−0.11
4	−1.42	3.43	0	0	−0.14	0.14	0	0	0.08	−0.08
5	−1.01	3.88	−0.03	0.03	−0.16	0.16	0.02	−0.02	0.06	−0.06
6	−0.62	4.31	−0.06	0.06	−0.16	0.16	0.03	−0.03	0.05	−0.05
7	−0.24	4.73	−0.09	0.09	−0.16	0.16	0.04	−0.04	0.03	−0.03
8	0.13	5.15	−0.11	0.11	−0.15	0.15	0.03	−0.03	0.03	−0.03
9	0.51	5.57	−0.12	0.12	−0.14	0.14	0.03	−0.03	0.02	−0.02
10	0.91	6.01	−0.13	0.13	−0.12	0.12	0.03	−0.03	0.01	−0.01
11	1.34	6.48	−0.12	0.12	−0.09	0.09	0.02	−0.02	0.01	−0.01
12	1.82	7.02	−0.11	0.11	−0.07	0.07	0.02	−0.02	0.01	−0.01
13	2.42	7.68	−0.09	0.09	−0.05	0.05	0.01	−0.01	0	0
14	3.28	8.63	−0.06	0.06	−0.03	0.03	0.01	−0.01	0	0
15	4.52	10	0	0	0	0	0	0	0	0

## Data Availability

The data presented in this study are available on request from the corresponding author. The data are not publicly available due to ethical restrictions.

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
