# Peer review of "Rasch Model of the COVID-19 Symptom Checklist—A Psychometric Validation Study"

_viruses, 2021, doi:10.3390/v13091762_

Round 1

Reviewer 1 Report

I would recommend the following two observations:

1). To make the narrative more crispy to a physicians' audience. I.e., why should a clinician be interested in this paper? What "added value" does the psychometric analysis "brings to the table" from a clinical viewpoint? What is novel about the study besides that only one study from Italy (reference #6) has performed something similar? Why a clinician should care about the Rasch models? Are there other similar psychometric models that perform better, worse, or equally well?  

2). To increase the resolution of the figures. 

Reviewer 2 Report

  • Please remove, from intro and discussion, “for the first time” we did this study, since it is not possible to assess this claim.
  • Move the last sentence of intro “Construct validity refers to the degree …” to the first paragraph of discussion
  • In line 89, after “…..very poor diagnostic power”, add some more sentences emphasizing the importance of molecular screening and diagnostic tests. A relevant study showed that use of questionnaires will not identify all infected patients and screening and diagnostic tests are necessary. Please cite and use this study to add more to the intro.

"Testing for COVID-19 in dental offices: mechanism of action, application and interpretation of laboratory and point-of-care screening tests." The Journal of the American Dental Association (2021).

  • Was there any sample size calculation?
  • Add statistical analysis section to methods and describe all the methods then have been implemented
  • The authors do not clearly mention how the construct validity and internal consistency of the questionnaire was. Please clarify in abstract and conclusion
  • Was the English version of the questionnaire used or Austrian version? If English version was used, how the translation could affect the results?

Author Response

This manuscript is a resubmission of an earlier submission. The following is a list of the peer review reports and author responses from that submission.

Round 1

Reviewer 1 Report

This is a timely and well-designed study on the validation of an online self-reported COVID-19 symptom checklist using the Rasch model. Fit to the Rasch model was examined in a sample of over 1600 Austrian citizens. Overall, the paper is well written and the method appropriate and well described. Nevertheless, it is not clear if this was an instrument development or a psychometric validation study. I have some comments and points that need further clarification:

Introduction

  • What I have missed is a short description/introduction regarding the recent evidence/availability of COVID-19 assessments/symptom checklists. You only say that psychometric properties of these instruments are scarce. But are there already instruments that have been validated? What were the results? Why did you had to develop a new checklist? Why did you include symptoms as e.g. cough and sore throat despite it had very poor diagnostic power in the recent Cochrane review?
  • You present the Rasch analysis/model very nicely, but I think that this method is not so novel in health sciences that it needs this much explanation. I would rather focus on the need and the gap of the study.
  • Regarding the aim of your study, you should clearly specify what you mean by “fundamental principles of measurement”. And what do you mean with “measurement precision”? Please explain more in detail. What was the purpose of your study and of using the Rasch model (i.e. to validate content validity, construct validity, reliability/internal consistency, responsiveness). See COSMIN taxonomy: Mokkink, L.B., Terwee, C.B., Patrick, D.L., Alonso, J., Stratford, P.W., Knol, D.L., Bouter, L.M. & de Vet, H.C. (2010) The COSMIN study reached international consensus on taxonomy, terminology, and definitions of measurement properties for health-related patient-reported outcomes. J Clin Epidemiol, 63(7), 737-45.
  • Please select/consider a reporting guideline for instrument validation studies. See also: STREINER D.L. & KOTTNER J . ( 2 0 1 4 ) Recommendations for reporting the results of studies of instrument and scale development and testing. Journal of Advanced Nursing 70(9), 1970–1979. doi: 10.1111/jan.12402. They say that “Newly developed scores or instruments should not be validated using the data set with which the instrument was developed. This usually leads to overly optimistic results. Therefore, please state the aim of your study more clearly.

Materials and methods

  • Please provide more information on how you reached/recruited the participants. What were the inclusion/exclusion criteria? How did you ensure informed consent? What were the data security measures regarding the online platform used for the study?
  • I do not understand when the participants filled in the questionnaire. Was it on a regular basis or if they experienced any symptoms, or just randomly? Please provide more information on the measurement timepoints and the intervals between the measurements.
  • How did you conduct self-reported comorbidities?
  • The Rasch analysis/fit to the Rasch model is very well described.
  • Why did you not consider to include qualitative research methods (e.g. cognitive interviews) when developing the instrument?

Results

  • The first sentence is unclear for me. How did you include participants from 0 to 9 years, i.e. without reading/writing abilities? For children below 18 years, did you had a consent of their parents? How do I have to understand these age groups (i.e. 0 to 9 years to ≥90years)? Additionally, for the age, you could provide median/IQR.
  • What about the response scale structure? Couldn’t you also look at the item characteristic curves (ICC)?

Discussion

  • What are the implications for clinical practice and research? Please elaborate more in detail.
  • What are the limitations of your study? Please be more clear about these points/decisions, which should/could have been done differently.

Reviewer 2 Report

The manuscript presents the evaluation of the analysis of a COVID-19 symptom questionnaire. The analysis methodology using the Rasch model allows evaluating the fit to a probability model and provides interpretable results, this is novel and original. However, the manuscript  requires extensive editing that allows understanding the origin of the problem, the results, the importance of the analysis, elements of the discussion. 

Here are some suggestion. 

Abstract 

L.18: specify the specific problem more directly in relation to COVID-19, avoid the generalization "Inaccurate scales in self-reported instruments" 

L.19: change “advanced psychometric modeling to determine if the fundamental principles of measurement” for a phrase that is much better understood by the reader 

Introduction. 

  L.38.Beforeintroducing the problem, it is better to talk about the current situation in relation to the measuring instruments and what is it, how they are used. 

L.43: What is considered a self-report instrument? What differences are there in the symptom observation scales? 

L.44: Why is oncology mentioned? 

L49 to 72: The purpose of the manuscript is not to use the Rasch model, the model is used in the method to elucidate a problem, not to put it in the center of the introduction. The problem as mentioned is the COVID-19 measurement instruments (of which nothing or almost nothing is explained), the problem is not the solution through the Rasch model. I suggest removing this entire section, inserting it into the shorthand method, and dedicating more space to the research problem.  

Here it should be explained what the fundamental principles of measurement are,and how it relates to what is being addressed. Before saying that the Rasch model respects the fundamental principles of measurement, it is better to explain why previous analyzes of the instruments to measure COVI-19 have not respected them. 

L 74: Quote 4 does not match the sentence. 

L 74-6: this is the center of the problem although it is not explained or argued.   

Materials and Methods: 

L 88: the study design is not the analysis, what kind of design is this? 

L 90 –94: Make an ethics section and put this content. 

L 96-99: this explanation should be partially present in the introduction. 

L 100-102: This is analysis put under a new heading Rash analysis. 

L 109-114: put under the heading ethics. 

L115. Put all procedures that involve rasch analysis in the Rasch analysis heading.  

L 115 and following. Explain clearly and briefly what is the objective of each step of the rasch analysis. (model Fit, Unidimensionaliy etc ..) 

L 1157; include this section in a larger section or explain what can be observed. 

L 158-164: explain how to change logits to the desired metric unit. 

Discussion. 

 The discussion requires a broad edition in which scientific norms are respected. What is expected is that an interpretation of the results is presented in relation to the scientific evidence. Instead it seems to be a presentation of the results without any dialogue with prior knowledge. 

Conclusions. 

The conclusions should reflect the final interpretation of the results and the progress that the paper represents.